biomaterials

bFGF, sustained release, self-assembling peptide, porous nHA/PA66, bone mesenchymal stem cells

**Authors for correspondence:**
Bo Qiao
e-mail: qiaobo1985@163.com
Dianming Jiang
e-mail: jdm571026@vip.163.com

†These authors contributed equally to this work.

# Controlled release of basic fibroblast growth factor from a peptide biomaterial for bone regeneration

WeiKang Zhao[1,2,†], Yuling Li[3,†], Ao Zhou[2], Xiaojun Chen[4], Kai Li[1,2], Sinan Chen[1,2], Bo Qiao[1] and Dianming Jiang[1,2]

[1]Department of Orthopaedics, The First Affiliated Hospital of Chongqing Medical University, No. 1 Youyi Road, Chongqing, Yuzhong District 400016, People's Republic of China
[2]Department of Orthopaedics, Third Affiliated Hospital of Chongqing Medical University, No. 1 Shuanghu Road, Chongqing City, Yubei District 401120, People's Republic of China
[3]Department of Orthopaedics, Affiliated Hospital of North Sichuan Medical College, No. 63 Wenhua Road, Nanchong City, Sichuan Province 637000, People's Republic of China
[4]Department of Orthopaedics, Hospital (T.C.M) Affiliated to Southwest Medical University, No. 182 Chunhui Road, Luzhou City, Sichuan Province, 646000, People's Republic of China

WKZ, 0000-0002-5594-213X

Self-assembled peptide scaffolds based on D-RADA16 are an important matrix for controlled drug release and three-dimensional cell culture. In this work, D-RADA16 peptide hydrogels were coated on artificial bone composed of nano-hydroxyapatite/polyamide 66 (nHA/PA66) to obtain a porous drug-releasing structure for treating bone defects. The developed materials were characterized via transmission electron microscopy and scanning electron microscopy. The proliferation and adhesion of bone mesenchymal stem cells (BMSCs) were examined by confocal laser microscopy and CCK-8 experiments. The osteogenic ability of the porous materials towards bone BMSCs was examined *in vitro* by staining with Alizarin Red S and alkaline phosphatase, and bioactivity was evaluated *in vivo*. The results revealed that nHA/PA66/D-RADA16/bFGF reduces the degradation rate of D-RADA16 hydrogels and prolongs sustained release of bFGF, which would promote BMSCs proliferation, adhesion and osteogenesis *in vitro* and bone repair *in vivo*. Thus, it deserves more attention and is worthy of further research.

## 1. Introduction

Large-area bone defects are frequently encountered in the clinic and their delayed healing or even lack of healing is associated with a

serious risk of mortality [1]. Artificial bone grafting is a promising therapeutic alternative to autologous bone grafting for the repair of bone defects [2,3].

In the field of bone tissue replacement and repair, biomaterials composed of bioactive inorganic compounds (such as nano-hydroxyapatite (nHA)) and macromolecular compounds (such as polyamide 66 (PA66)) can provide better mechanical properties and ductility. Specifically, nHA is the main mineral component of bones. Therefore, nHA is extremely biocompatible and does not promote inflammatory responses. At the same time, PA66 mimics the collagen component in bones, making the material more ductile [4–6]. Porous nHA/PA66 composites are commonly used in the clinic as artificial bone materials owing to their biocompatibility and suitable mechanical properties [7,8]. However, the lack of biological activity of these materials limits their performance in clinical applications. Coating with nanofibre materials could afford porous nHA/PA66 composites with controlled release properties and the appropriate bioactivity.

Previous studies have demonstrated that the D-RADA16 peptide [9] can be used for drug delivery [8,10]. Self-assembling peptide hydrogels composed of D-amino acids are more resistant to enzymatic degradation than those composed of L-amino acids [11], and the controlled release of a peptide depends on its stable three-dimensional network structure. Furthermore, the D-RADA16 scaffolds were reported to significantly promote the delivery of both oxygen and soluble signalling molecules [12,13]. Other studies also reported that D-RADA16 can accelerate the process of wound repair following injury by delivering drugs to the damage site [14,15]. However, D-RADA16 will be rapidly degraded by proteases *in vivo* and sudden release of the loaded drugs [16]. The exposed surface area affects the degradation rate of the material and the slow release efficiency of the drug [12]. Therefore, this study investigated the possibility of introducing self-assembled D-RADA16 hydrogel scaffolds into the pores of porous nHA/PA66 composites. The underlying hypothesis was that the D-RADA16 peptide structure could be stabilized by the pores of nHA/PA66 composites and permit sustained release of basic fibroblast growth factor (bFGF) incorporated into the hydrogel. bFGF stimulates neovascularization during the early stages of healing and promotes the proliferation and differentiation of bone mesenchymal stem cells (BMSCs), thus holding great potential in promoting the healing of bone defects [17,18]. However, bFGF has a short half-life *in vivo*, which leads to insufficient stimulation of local cells [12]. Furthermore, burst release can easily lead to excessive bFGF concentrations, which may interfere with healing [15,19]. Previous experiments also revealed that D-RADA16 hydrogel and bFGF-conjugated D-RADA16 hydrogel did not display differences in terms of promoting bone healing *in vivo* [9]. Therefore, the key to successful treatment may be the long-term sustained release of bFGF in damaged bone [20].

In the present study, we examined the controlled release properties of nHA/PA66/D-RADA16 scaffolds. Using both *in vivo* and *in vitro* experiments, we investigated the effects of the scaffolds on promoting cell proliferation, differentiation and osteogenesis with the aim of developing a potential therapeutic strategy for patients with bone defects.

# 2. Material and methods

## 2.1. Preparation of peptide and bFGF

Self-assembling peptide D-RADA16 (Ac-RADARADARADARADA-CONH2, 1712.78 g mol$^{-1}$, purity greater than 98.58%) was synthesized by Shanghai Biotech Bioscience & Technology (Shanghai, People's Republic of China). And the N-terminus and C-terminus of the peptide were protected by acetyl and amino groups, respectively. The peptide D-RADA16 sequence contained all D-amino acids. Solutions of the peptides were prepared at concentrations of 5 mg ml$^{-1}$ in water (18 MΩ cm; Millipore Milli-Q System, Billerica, MA, USA) and stored at 4°C before use. The bFGF solution, was purchased from Guizhou Dida Biological Technology Co., Ltd (Guiyang, Guizhou Province, People's Republic of China), was stored at −20°C and was used without further purification.

Lyophilized peptide powder was dissolved in deionized water to obtain a 10 mg ml$^{-1}$ (w/v) peptide stock solution at 25°C. Peptide hydrogels in each group were mixed well at 25°C. In group one, combining 10 mg ml$^{-1}$ (w/v) D-RADA16 peptide solution with PBS (pH 7.4) by the volume ratio of 1 : 1, and with the stimulation of ionic solution PBS, 5 mg ml$^{-1}$ D-RADA16 peptide solution would undergo self-assembly for 24 h in order to produce D-RADA16 peptide hydrogel. While in the bFGF group, 50 µg ml$^{-1}$ (w/v) bFGF solution was prepared by dissolving bFGF powder into PBS (pH 7.4). And then 10 mg ml$^{-1}$ (w/v) D-RADA16 peptide solution was combined well with 50 µg ml$^{-1}$ (w/v) bFGF by the volume ratio of 1 : 1.

## 2.2. Preparation of nHA/PA66/D-RADA16 and mechanics test

nHA, the main inorganic component of human and animal bones, and PA66 materials were purchased from Nanotechnology Co., Ltd (Chengdu, People's Republic of China). The synthesis of materials was completed at the Nanotechnology company. nHA/PA66 composite slurry was then prepared in accordance with methodology described previously [5]. Foam agent, which would turn to a gaseous phase at high temperature (1 : 50) was added to the composite PA66 slurry at 70°C. Then, the PA66/foam was added to the nHA slurry and stirred constantly for 2 h. Throughout this paper, the nHA/PA66 composites used contained 40% (by weight) of nHA. The composite slurry was then allowed to set for 4 h at room temperature, and the bubbles were completely removed. The slurry was then cast onto a plate and the composites heated to 300°C for 1 h. The resultant materials were then washed repeatedly with deionized water. The preparation of several specifications of porous nHA/PA66 is completed. The calculation formula of the material porosity is as follows:

$$\text{porosity of porous bone filling material} = \left(1 - \frac{\text{porous bone filling material density}}{\text{non-porous solid material density}}\right) \times 100\%.$$

The prepared porous nHA/PA66 composites were completely immersed in D-RADA16/bFGF solution (5 mg ml$^{-1}$ peptides is the optimum coating concentration, which can form a stable three-dimensional structure without agglomeration, as shown in the electronic supplementary material) at room temperature by dissolving lyophilized peptide powder in PBS (pH 7.4) for 24 h to make the nHA/PA66/D-RADA16/bFGF composite for further studies. Mechanical tests were also carried out on the two composites using 15 × 10 × 10 mm samples. These tests showed that pressing speed was 5 mm min$^{-1}$ and compression height ratio was 50%.

## 2.3. SEM and TEM test

The surface morphologies of the nHA/PA66/D-RADA16 and nHA/PA66 composites were characterized via scanning electron microscopy (SEM) (Nova NanoSEM 400, FEI, Hillsboro, OR, USA). Transmission electron microscopy (TEM) (Philips Tecnai G2 F20, FEI Company, Hillsboro, OR, USA) was employed to image the nanofibre of D-RADA16. The D-RADA16 composite used in our study was tested as follows. D-RADA16 peptide powder was dissolved in PBS for 48 h at room temperature at a concentration of 5 mg ml$^{-1}$. Next, 10 ml of peptide solution was loaded onto a copper grid. Redundant solution was removed, and 10 ml of uranyl acetate was used to dye the peptide solution for 30 s; the solution was then dried for 12 h in a desiccator. Finally, TEM (Philips Tecnai G2 F20, FEI Company, Hillsboro, OR, USA) was used to take images. Image-Pro Plus was employed to measure the macropore size of nHA/PA66 and the length as well as width of nanofibres ($n = 12$).

## 2.4. Peptide degradation by proteinase K

Proteinase K solution (RayBiotech, Norcross, GA, USA) (0.05 mg ml$^{-1}$) was added in D-RADA16 and nHA/PA66/D-RADA16 groups, respectively. All samples were incubated at 37°C, and 400 µl aliquots were removed at various time points (1, 2, 4, 6, 8, 12, 24 and 48 h). The enzymatic reaction was terminated by the addition of phenylmethanesulfonyl fluoride (working concentration: 0.1 to 1 mM) and the concentration of the intact peptide were analysed by HPLC (LCMS-2020; Shimadzu, Kyoto, Japan) under the following conditions: Venusil XBP C18, 5 µm, 4.6 × 50 mm (Agela, Tianjin, China); sample injection volume: 100 µl; gradient elution: A, 0.035% trifluoroacetic acid (TFA) in water; and B, 0.035% TFA in acetonitrile/water (80 : 20, vol/vol). A gradient of 10–100% B in 10 min was conducted. The experiment was conducted in three replicates.

## 2.5. bFGF release

The bFGF release from three different materials, namely, D-RADA16 scaffolds, nHA/PA66 scaffolds and nHA/PA66/D-RADA16 scaffolds, without BMSCs was investigated. bFGF solution was added to the D-RADA16 solution such that each material contained 10 ng of bFGF. Each 24-multiwell plate (NEST Biotechnology Co., Ltd, Rahway, NJ, USA) contains 20 µl of mixed solution and then was stimulated to self-assemble after adding 180 µl of PBS (7.4 pH) in each well for 60 min. Standard DMEM-F12 medium was employed for the experiments (500 µl per well). bFGF enzyme-linked immunosorbent

assay kits (RayBiotech, Norcross, GA, USA) were used to analyse the bFGF concentration. A total of 40 µl of supernatant of the mixed solution in each well was taken away at the time point of 1, 2, 4, 8, 24, 72 and 168 h, while the same dose of fresh PBS solution was added into each well slowly and gently. These supernatant samples were stored in a −80°C refrigerator for the following ELISA determination. The amounts of bFGF in each group were then obtained and the diffusion coefficients were calculated according to the unsteady state form of Fick's second law.

## 2.6. Cell culture

BMSCs were obtained from four-week-old female Sprague Dawley (SD) rats using a protocol approved by the Ethics Committee of the First Affiliated Hospital of Chongqing Medical University (permit no. 2014-201058). All cells were cultured in complete DMEM containing 10% foetal bovine serum (FBS, Invitrogen/Thermo Fisher, Waltham, MA) with penicillin (100 units ml$^{-1}$) and streptomycin (100 µg ml$^{-1}$) at 37°C under 5% $CO_2$. Unless otherwise indicated, all other reagents were obtained from Sigma-Aldrich (St Louis, MO) or Thermo Fisher Scientific (Waltham, MA). All cells used in experiments are within five passages. The cell density in each well was $3 \times 10^4$ cells cm$^{-2}$. The osteogenic induction medium was replaced every 3 days.

## 2.7. CLS test

In this experiment, $1 \times 1 \times 0.1$ cm specimens of the different materials were prepared. BMSCs were cultured with different groups (glass control, D-RADA16, nHA/PA66 and nHA/PA66/D-RADA16 scaffolds) on a 24-well plate for 12 h. The cell density in each well was calculated to be approximately $3 \times 10^4$ cells cm$^{-2}$ using cell counting plates.

The cells were fixed in 4% paraformaldehyde for 15 min, Triton X-100 was added as a detergent, and the cells were stained with phalloidin (50 µg ml$^{-1}$; Cyagen, Guangzhou, China) for 30 min and then removed. Next, the cells were stained with 4′,6-diamidino-2-phenylindole (0.1 mg ml$^{-1}$; Cyagen, Guangzhou, China) for 15 min and imaged using confocal laser microscopy (CLS, Nikon, Tokyo, Japan). Finally, the cytoskeleton morphology was examined using inverted fluorescence microscopy (Leica SP8, Wetzlar, Germany) and Image-Pro Plus software. Cellular F-actin and bFGF were estimated from the aspect ratio and number of cell branching points. To evaluate the deformation, the cell roundness was measured using the Image-Pro Plus software (Media Cybernetics, Inc., USA). The cell roundness is expressed as the ratio of the average radius of curvature of the edge or corners to the radius of curvature of the maximum inscribed ball [12].

## 2.8. CCK-8 and EdU staining

For quantitative analysis, BMSCs ($2.5 \times 10^3$ cells well$^{-1}$) were seeded onto different groups (glass control, nHA/PA66/D-RADA16, nHA/PA66/bFGF and nHA/PA66/D-RADA16/bFGF). At defined time points of 6, 12, 24 and 72 h after seeding, the cell proliferation was assessed using the Cell Counting Kit-8 (CCK-8; Dojindo, Kumamoto, Japan); the optical density of the medium was determined at 460 nm using a plate reader (Thermo Fisher Scientific Co., Ltd, Waltham, MA, USA). Each experiment was performed in triplicate and repeated three times.

After BMSCs co-cultured with different groups (glass control, nHA/PA66/D-RADA16, nHA/PA66/bFGF and nHA/PA66/D-RADA16/bFGF) for 72 h, the 10 µM 5-ethynyl-2-deoxyuridine (EdU; EdU Click-iT® Imaging Kit, Molecular Probes, Invitrogen) was added to culture media and cells incubated for a further 2 h. BMSCs were fixed in 2% paraformaldehyde for 30 min at room temperature followed by four successive washes with $1 \times$ PBS. Lens explants were permeabilized in 0.5% Triton-X100 (VWR, IL, USA) in PBS for 20 min, followed by three successive washes in PBS and two washes in 3% BSA/PBS for 5 min and 10 min, respectively. AlexaFluor 488-azide was used to stain samples and immunofluorescence microscopy was employed for measurement.

## 2.9. Alizarin Red staining and ALP determination

After BMSCs co-cultured with different groups (glass control, nHA/PA66/D-RADA16, nHA/PA66/bFGF and nHA/PA66/D-RADA16/bFGF) for 21 days, the materials were fixed with paraformaldehyde(4%) for 15 min and treated with Alizarin Red S (1%; Sigma-Aldrich, St Louis, MO, USA) to stain the calcium deposits and measure the osteogenic differentiation of the BMSCs. The samples were imaged using an optical microscope (Carl Zeiss, Jena, Germany) and cetylpyridinium

chloride (10%; Cyagen, Guangzhou, China) was used to quantitatively analyse the calcium nodules. Samples were analysed at a wavelength of 620 nm.

The early osteogenic differentiation of the BMSCs was determined via alkaline phosphatase (ALP) staining, and the activity, using ALP Assay Kit (Abcam, Burlingame, CA) and following the manufacturer's protocol, was measured at 7 days after co-cultured. The BMSCs in each groups (glass control, nHA/PA66/D-RADA16, nHA/PA66/bFGF and nHA/PA66/D-RADA16/bFGF) were fixed with 4% paraformaldehyde for 15 min and then stained for 0.5 h with the prepared ALP solution (Sigma-Aldrich, St Louis, Missouri, USA) for staining. After three times PBS washing, an optical microscope (Carl Zeiss, Jena, Germany) was used to image samples. At the same time, the ALP activity was quantitatively analysed by a microplate reader (Multiskan, Thermo Fisher Scientific, MA, USA) at a wavelength of 490 nm.

## 2.10. *In vivo* experiments

Twenty-four four-week-old female SD rats each weighing 200–240 g were obtained from the Laboratory Animal Center in Chongqing Medical University. The protocol was approved by the Ethics Committee of the First Affiliated Hospital of Chongqing Medical University Reference (IACUC no. 2016-059). The rats were randomly divided into two groups of 12 animals, such that there were six animals for each group at each time point. The rats were anaesthetized using chloral hydrate (10%, 5.0–7.5 ml kg$^{-1}$), the left distal femoral condyle of each rat was exposed, and a bone defect with a diameter of 2.5 mm and depth of 3 mm was created and the right distal femoral condyle was prepared as a control group (without filling materials). The defects were filled with either nHA/PA66/bFGF (5 mg ml$^{-1}$) or nHA/PA66/D-RADA16/ bFGF (5 mg ml$^{-1}$). After finishing surgery, penicillin was injected to prevent potential infection. After 8 or 12 weeks feeding in The Laboratory Animal Center, the rats were euthanized using $CO_2$ and the femoral condyles were harvested for analysis via micro-computed tomography (micro-CT).

## 2.11. Micro-CT analysis

The femoral condyle samples were subjected to micro-CT analysis (Viva CT40, Scanco Medical AG, Bassersdorf, Switzerland) at 70 kVp, 114 µA, a voxel size of 7.0 µm and a slice thickness of 0.01 mm. The average scanning time was approximately 45 min. Three-dimensional images of the new bone structure in the defect area were computationally reconstructed. Quantitative measurements of the new bone tissues were used to calculate the bone volume per tissue volume (BV/TV), trabecular separation (Tb.Sp) and trabecular thickness (Tb.Th). Trabecular number (Tb.N) and trabecular thickness (Tb.Th) are two key indicators to describe the number and thickness of trabecular bone, and Tb.Sp is used to measure the separation of trabecular bone.

## 2.12. Histology

The harvested specimens (femoral condyle) were fixed with 4% paraformaldehyde solution. After decalcification with ethylene diamine tetra acetic acid, the samples were dehydrated using an ethanol series, embedded in paraffin, sliced to a thickness of 6 µm, and stained with haematoxylin and eosin (H&E).

## 2.13. Statistical analysis

SPSS v. 20.0 (SPSS Inc., Chicago, IL, USA) was used for statistical analysis. Data are expressed as the mean ± s.d., and differences between groups were examined via the analysis of variance (ANOVA) method. Differences were considered statistically significant at $p < 0.05$.

# 3. Results

## 3.1. SEM, TEM and mechanical test

TEM analysis demonstrated that D-RADA16 successfully self-assembled to form network structures composed of interwoven nanofibres (figure 1*a*). Results of TEM showed that the D-RADA16 peptide nanofibres were 10–36 nm in width and 147–602 nm in length.

Results of SEM showed that the nHA/PA66 macropore size was approximately 635.68 ± 122.50 µm with a porosity of 75.47 ± 5.31% (figure 1*b,c*). As results of nHA/PA66/D-RADA16/bFGF SEM

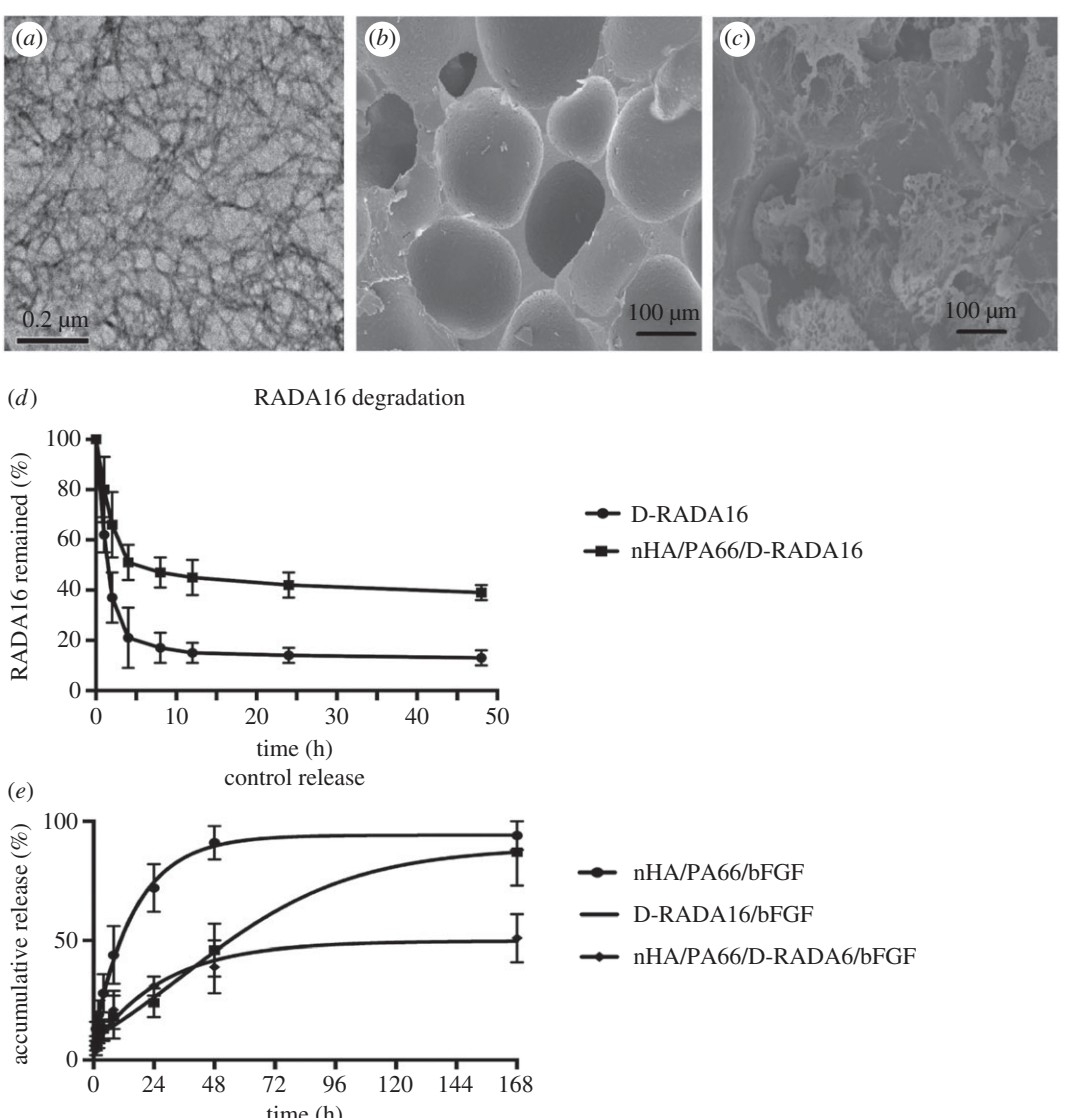

**Figure 1.** The results of material characterization, D-RADA16 degradation and bFGF sustained release. (*a*–*c*) Materials characterization and (*d*) degradation, (*e*) bFGF sustained release. (*a*) TEM image showing the microstructure of a D-RADA16 hydrogel. (*b*,*c*) SEM images showing the microstructures of (*b*) nHA/PA66 and (*c*) nHA/PA66/D-RADA16/bFGF composites. (*d*) Degradation of D-RADA16 using proteinase K test ($n = 3$). (*e*) bFGF release with proteinase K test ($n = 3$).

exhibited, we found that D-RADA16 forms a network structure and is distributed on the material surface; especially, it can aggregate in the void structure of nHA/PA66 material. These network structures composed the base of drug release. At the same time, the results of compressive strength showed that none of the physical parameters of nHA/PA66/D-RADA16 were significantly different from those of the nHA/PA66 scaffolds ($p > 0.05$) (electronic supplementary material, table S1).

## 3.2. D-RADA16 degradation

Proteinase K, a highly non-specific proteolytic enzyme, was used to examine the stability of D-RADA16 in the various composites. The concentration of each D-RADA16 was set to 0.2 mg ml$^{-1}$. And after 4 h, only 30–40% of the peptide remained in the control samples, whereas over 50% of the peptide still remained in the experimental samples. Similar results were observed after 24 and 48 h (figure 1*d*).

## 3.3. Controlled release of bFGF

The nHA/PA66/bFGF group exhibited rapid burst release (figure 1*e*), where approximately 90% of the encapsulated bFGF was released during the first 48 h. By contrast, the D-RADA16/bFGF group showed

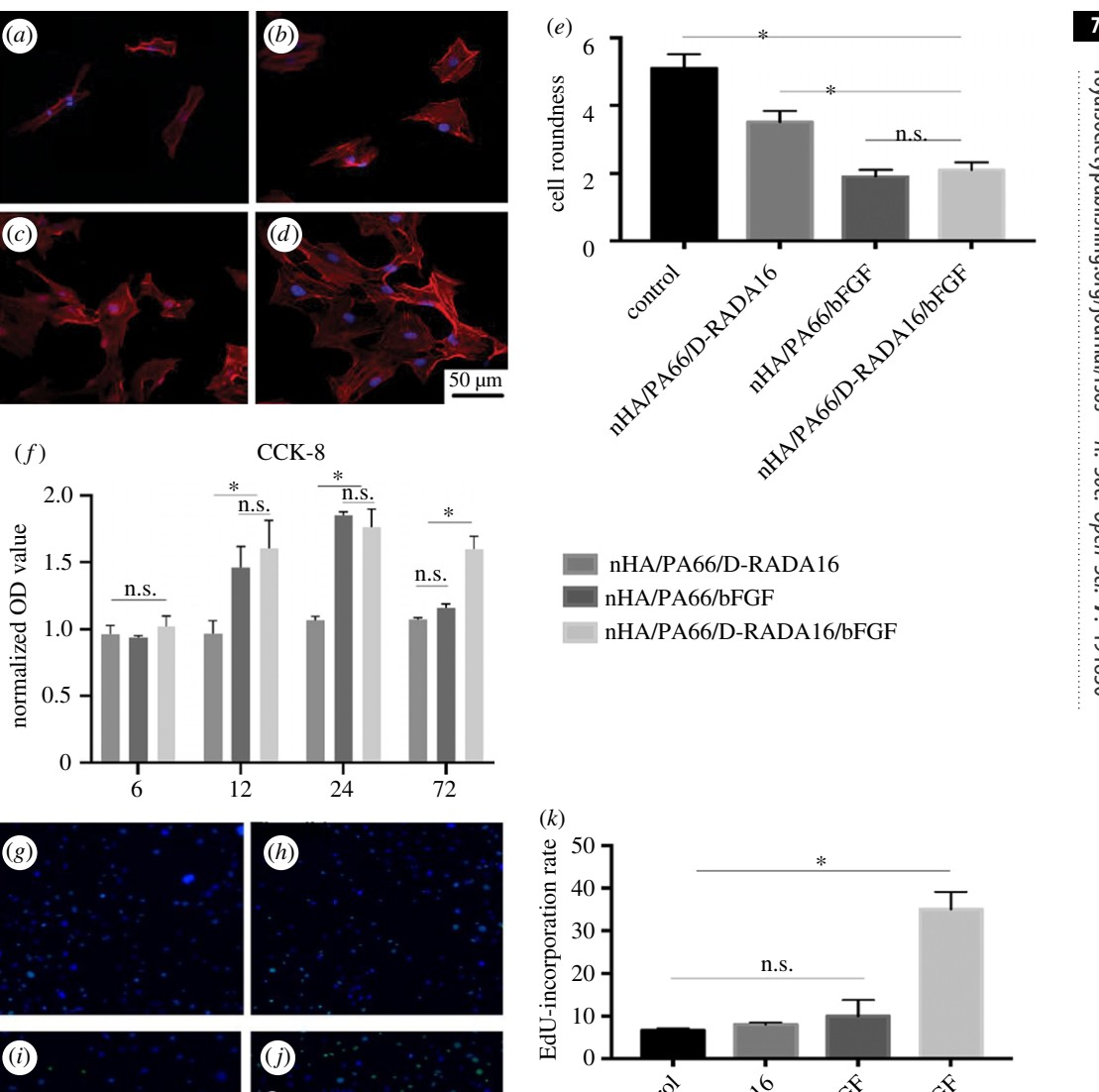

**Figure 2.** The results of CLS, CCK-8 and EdU staining. Influence of materials on (a–e) cell roundness, CCK-8 (f) and EdU staining (g–k). (a,g) Control group, (b,h) nHA/PA66/D-RADA16 group, (c,i) nHA/PA66/bFGF group, (d,j) nHA/PA66/D-RADA16/bFGF group, (e) Cell roundness test, cell roundness form factor ($f = 4\pi a/p^2$; a, area; p, perimeter) analysed by Image J (BMSCs, control: $n = 37$; nHA/PA66/D-RADA16: $n = 33$; nHA/PA66/bFGF: $n = 29$; nHA/PA66/D-RADA16/bFGF: $n = 39$) (f) CCK-8 test ($n = 3$), (k) EdU-incorporation rate ($n = 3$), (* $p < 0.05$).

that rapid burst release during the first 48 h was inhibited, but the residual bFGF amount was the non-significant difference ($p > 0.05$) between in the nHA/PA66/bFGF group. Furthermore, the nHA/PA66/D-RADA16/bFGF composite displayed consistent bFGF release without obvious burst release within 48 h, followed by slower bFGF release over 168 h ($p < 0.05$).

## 3.4. CLS

The BMSCs were imaged via CLS to identify the cell microfilaments with the phalloidin label (figure 2a–d). The cells cultured in the presence of the nHA/PA66/D-RADA16/bFGF composite exhibited a long and spindle-shaped morphology. Cytoskeletal staining and roundness statistics were used to evaluate the

occurrence of cellular deformation on the surfaces of the various materials. The results revealed that cellular deformation was reduced after culturing for 24 h in the presence of the bFGF-containing composites (figure 2e). Compared with the control group, the cell roundness in the nHA/PA66/bFGF and nHA/PA66/D-RADA16/bFGF groups was reduced by 61% and 58%, respectively. These results suggest that the controlled released of bFGF can reduce cellular deformation. Furthermore, the cell roundness in the nHA/PA66/D-RADA16 group was reduced by 31% compared with the control group (figure 2e).

## 3.5. Cell proliferation

The CCK-8 results indicated that there were no significant differences between each group at 6 h. As for later time points, bFGF could promote cell proliferation in nHA/PA66/bFGF and nHA/PA66/D-RADA16/bFGF groups (figure 2f). Specifically, the cell proliferation in nHA/PA66/bFGF and nHA/PA66/D-RADA16/bFGF groups were significantly higher than groups (control and nHA/PA66/D-RADA16) without bFGF at 12 and 24 h. This effect was more persistent in the nHA/PA66/D-RADA16/bFGF group than in the nHA/PA66/bFGF group, and cell proliferation remained significantly increased after 72 h in the former case. We can conclude that bFGF-containing materials can significantly promote cell proliferation, while slow-release bFGF materials can extend the effect of cell proliferation to a minimum of 72 h. According to the sustained-release experiment, nHA/PA66/bFGF would be expected to release approximately 70% of the loaded bFGF within 24 h, and the rapid loss of bioactivity for bFGF may explain the higher levels of cell proliferation observed in the case of nHA/PA66/D-RADA16/bFGF [20]. EdU staining reveals a consistent result that cell proliferation in nHA/PA66/bFGF group was not higher than glass control and nHA/PA66/D-RADA16 after 72 h (figure 2g–k).

## 3.6. Alizarin Red staining and ALP determination

BMSCs were cultured in the presence of various materials. Compared with control and nHA/PA66/D-RADA16 group, images from bFGF groups showed that a larger area of orange staining was observed (figure 3a–d), and results of the quantification of calcium nodule formation by the BMSCs exhibited that osteogenic abilities of BMSCs in nHA/PA66/D-RADA16/bFGF group were significantly higher than it in nHA/PA66/bFGF (figure 3e). What is more, the results of ALP staining and the activity exhibited the same tendency (figure 3f–j). In this research, bFGF was found to significantly promote the osteogenesis of the BMSCs, especially in the nHA/PA66/D-RADA16/bFGF group.

## 3.7. Micro-CT quantification and histology analysis

As shown in the three-dimensional image, At eight weeks after the surgery, a small amount of new bone formation was detected in the defects in the control group and nHA/PA66/bFGF group; however, defects in nHA/PA66/D-RADA16/bFGF group have been significantly repaired by new bone tissue. As for 12 weeks, defected bones in the nHA/PA66/D-RADA16/bFGF group were almost repaired, and nHA/PA66/bFGF group did not show any improvement compared with the control group. And the histology results showed the same tendency with the three-dimensional image. We use micro-CT to quantitatively detect changes in bone mass at the defect site. Comparison of the micro-CT images at the two time points (8 and 12 weeks) revealed improved repair of the bone defects in the group treated with nHA/PA66/D-RADA16/bFGF (figure 4). Analysis of the quantitative parameters (BV/TV, Tb.N, Tb.Th and Tb.Sp) measured via micro-CT demonstrated a significantly greater volume of regenerating bone in the nHA/PA66/D-RADA16/bFGF group ($p < 0.05$), which was consistent with the BV/TV results ($p < 0.05$). The nHA/PA66/D-RADA16/bFGF group also exhibited a significantly higher Tb.N ($p < 0.05$) and lower Tb.Sp of new bone ($p < 0.05$) compared with the nHA/PA66/bFGF group and negative control group.

# 4. Discussion

## 4.1. Material characteristics and controlled release

It is difficult to repair some bone defects, particularly load-bearing or large bone defects, which may cause non-union and thus cause an effect upon a patient's quality of life [4,21]. Bones do have the potential to regenerate; however, bone induction and conduction for the recovery of large or load-bearing bone defects require bone grafts, which are supported mechanically [3,5]. Allogeneic bone

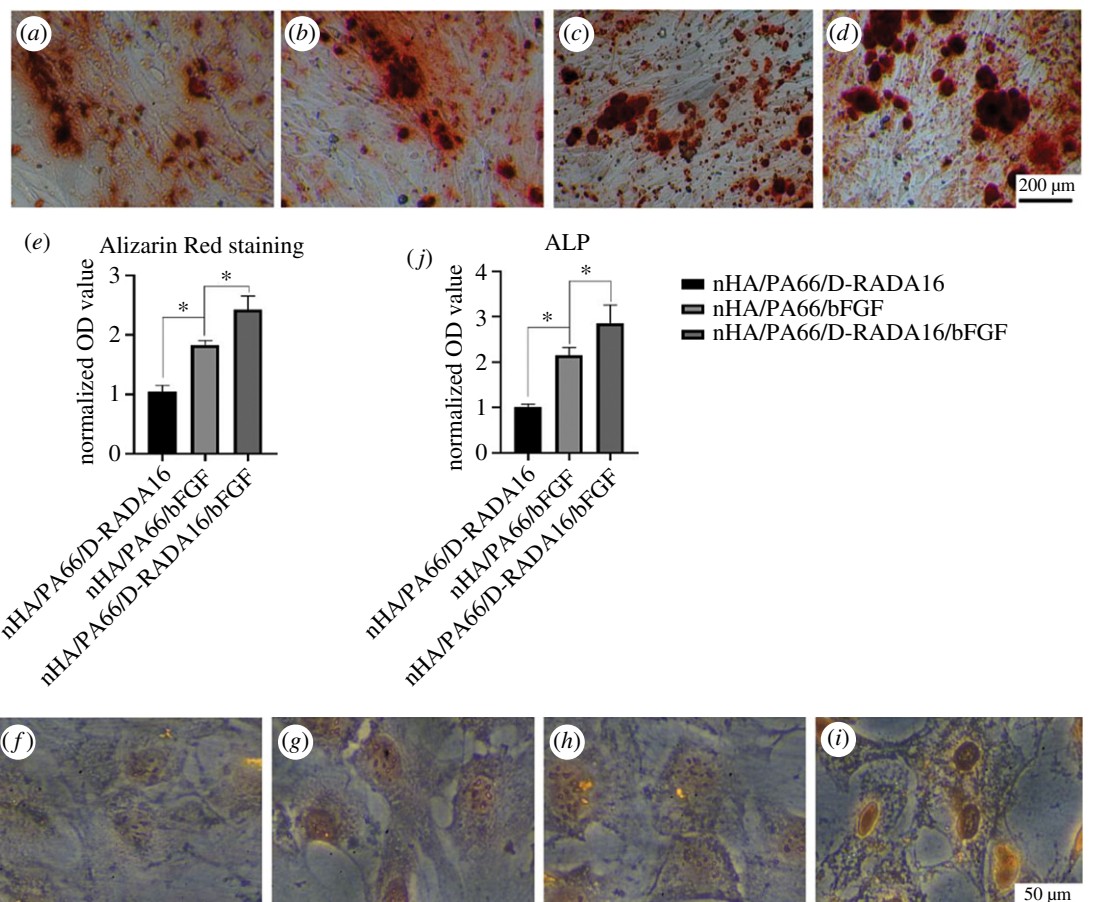

**Figure 3.** Alizarin Red S and ALP. Optical microscopy images (×100) showing the results of staining with (a–d) Alizarin Red S and (f–i) ALP. (a,f) Control group, (b,g) nHA/PA66/D-RADA16 group, (c,h) nHA/PA66/bFGF group, (d,i) nHA/PA66/D-RADA16/bFGF group. Quantitative analysis of (e) Alizarin Red S and (j) ALP activity test. OD values for each sample (n = 3, *p < 0.05).

grafts (e.g. demineralized bone matrix (DBM)) serve as an alternative to repair bone defects and are limited by the suboptimal osteoactivity when compared with autologous grafts, the risk of disease transmission and immune rejection. As to synthetic bone repair material scaffolds, they are mainly divided into polymers, bioceramics, metal materials, bioglass and composite materials [4]. However, there are still some disadvantages (e.g. low osteoinductivity and osteoconductivity, the difficulty in degradation or even non-degradation) for their application in bone regeneration [6]. nHA materials can provide good mechanical support, but the ductility is insufficient [4]. Calcium sulfate ($CaSO_4$) as a bone substitute was reported in 1892. However, the resorption of $CaSO_4$ is faster than the rate of new bone deposition [22], and thus, it is rather unsuitable as a material to support early functional rehabilitation [23]. Thus, this porous nHA/PA66 consists of nHA and PA66, and exhibits the features of good biocompatibility, osteoconductive and mechanical behaviour. Furthermore, this experiment has shown that nHA/PA66 not only satisfies the requirements to act as a substitute for natural bone but also has similar biomechanics (electronic supplementary material, table S1) (for example, compressive strength approximate from 2.5 to 4.5 Mpa for human cancellous bone [4,24]) and structural to human cancellous bone (electronic supplementary material, table S1) (for example, 65–85% porosity and pores from 100 to 200 µm for human cancellous bone [4–6,24,25]). This means that when nHA/PA66 with or without D-RADA16 is used as a bone filling material, it can not only provide mechanical support similar to cancellous bone but also avoid the stress shielding effect. A new biomaterial demands both preferable biomechanics and biological activity; however, nHA/PA66 composite has insufficient bioactivity compared with other biomaterials [3,21,22].

In this work, D-RADA16 was used to prepare nHA/PA66/D-RADA16 composites with controlled release ability. Three-dimensional scaffolds for biomedical applications should possess a porous micro-architecture with an appropriate pore size and interconnected pores. These properties are essential for

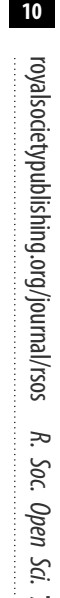

(*a*)  (*b*)  (*c*)  (*d*)  (*e*)  (*f*)   50 µm

(*g*)  (*h*)  (*i*)  (*j*)  (*k*)  (*l*)   200 µm

(*m*) BV

(*n*) BV/TV

control
nHA/PA66/bFGF
nHA/PA66/D16/bFGF

(*o*) TB.N

(*p*) Tb.Th

(*q*) Tb.Sp

(normal bone: BV: 14.37 ± 0.71; Tb.N: 6.12 ± 0.43; Tb.Th: 0.41 ± 0.09; Tb.Sp: 0.29 ± 0.04)

**Figure 4.** Osteogenic performance of various materials 8 and 12 weeks after implantation. (*a–f*) Reconstructed micro-CT images of the explanted femoral condyles and (*g–l*) histology (H&E stain) images of the explanted femoral condyles: (*a,g*) control group (eight weeks), (*b,h*) control group (12 weeks); (*c,i*) nHA/PA66/bFGF group (eight weeks), (*d,j*) nHA/PA66/bFGF group (12 weeks); (*e,k*) nHA/PA66/D-RADA16/bFGF (8 weeks), (*f,l*) nHA/PA66/D-RADA16/bFGF (12 weeks). (*m–q*) Osteogenic observation of various materials 8 weeks and 12 weeks after implantation. ($n = 6$, *$p < 0.05$).

the transport of nutrients and gases and the infusion of cells inside the scaffold. All of these factors are beneficial for tissue ingrowth and mechanical interlocking with the material. It was previously reported that pore sizes ranging from 40 to 135 µm led to osteoblast attachment to porous high-density polyethylene with enhanced viability.

The D-RADA16 nanofibres used in this study had a width of 6–30 nm and a length of 120–500 nm and were crosslinked into interwoven networks with a pore size ranging from 40 to 200 nm, which endowed the material with sustained-release ability [26]. The assembly of the peptide nanofibre hydrogel with a stable three-dimensional structure led to the slow and sustained release of bFGF. The secondary structure of D-RADA16 changed upon exposure to PBS, indicating that it is susceptible to ions. Self-assembling peptide nanofibre hydrogels can form three-dimensional pores with similar diameters, providing an ideal network structure for three-dimensional cell culture. At the same time, this pore size regime provides the potential for release of functional proteins [19,26,27]. The results revealed that the D-RADA16 was distributed both on the surfaces and in the gaps of the nHA/PA66 composites. Protein molecules in the interlaced nanofibre network structures of hydrogel scaffolds are physically hindered [21]. The highly interwoven nanofibre networks of hydrogels could limit movement of bFGF molecules [28,29]. The distinct bFGF release profiles observed for the composites could be ascribed to different D-RADA16 degradation rates in different groups. This could explain

why the D-RADA16/bFGF hydrogel displayed a similar bFGF release profile as nHA/PA66/D-RADA16/bFGF over the first 48 h. After 7 days, compared with D-RADA16 group, there was no significant difference in the amount of bFGF released by nHA/PA66/bFGF [21], probably due to the D-RADA16 complete degradation. These results demonstrate the excellent controlled release ability of the nHA/PA66/D-RADA16/bFGF composite. And most of the bFGF cannot move freely, which may be the origin of the sustained-release behaviour [9]. Upon subsequent degradation of the hydrogel, the encapsulated drug molecules can be released in a step-by-step manner over a prolonged period, which is especially important for tissue repair engineering [30,31] and promoting cell proliferation [32–34]. The adhesion, proliferation and differentiation of BMSCs all benefit from the structure of the D-RADA16 peptide scaffold. Several previous studies have demonstrated that hydrophobic domains of the scaffold and interactions between the nanostructure and cells or proteins could be responsible for these effects [7,35]. By cell morphology observed by immunofluorescence staining, we can conclude that bFGF affects early cell adhesion. After being co-cultured with different groups for 12 h, the morphology of BMSCs in the groups with bFGF was flat and the area large, indicating that the cell spreading is good. Many pseudopodium were observed in the cells, indicating that the cell extending is good, while the cells in the groups without bFGF were smaller with fewer pseudopods.

## 4.2. Cell proliferation

During sustained release, the biological activity of bFGF remains relatively stable. bFGF can accelerate the proliferation of BMSCs and promote their differentiation in the osteogenic direction [36,37]. The initial burst release of bFGF that occurred during the first 8 h may be attributable to the surface distribution of the protein [28,29]. The cell proliferation in all of the experimental groups was superior to that in the control group. In addition, ALP activity and calcium nodules deposition in the nHA/PA66/D-RADA16/bFGF group were higher than those in the control group.

During the first 24 h burst period, the release rates of the D-RADA16 group and the nHA/PA66/D-RADA16 group were similar and were much less than the nHA/PA66 group. We believe that D-RADA16 peptide can effectively reduce the burst release of drugs.

Compared with D-RADA16 group at 168 h, the significantly lower release from nHA/PA66/D-RADA16/bFGF was probably attributable to more stable three-dimensional structure formed by the D-RADA16 hydrogel in the nHA/PA66 pores. These experimental results were consistent with those observed upon enzymatic hydrolysis using proteinase K. Furthermore, we observed that the stability of the three-dimensional structure formed by the peptide hydrogel was not only the basis for its imitation of the extracellular matrix but also the reason for the significant improvement in the sustained-release ability of the material [31,38–40]. These results indicate that the sustained release of bFGF from the nHA/PA66/D-RADA16/bFGF hydrogel was more prolonged than that from the D-RADA16/bFGF hydrogel, leading to a lower local concentration of bFGF. No significant changes in the cell morphology, apoptosis or degeneration were observed, suggesting that the nHA/PA66/D-RADA16 material can be used as a material for bone tissue repair. In addition, owing to its sustained-release properties, this material may also be applicable to anti-infection and anti-tuberculosis treatments [11,41].

## 4.3. Osteogenesis *in vitro* and *vivo*

Bone consists of both organic (e.g. collagen I) and inorganic components (e.g. bony salts) [1]. Porous nHA/PA66 composites mimic a bony structure in the sense that nHA acts as a bony salt and PA66 acts as the porous structure to cancellous bone; however, lots of research showed that nHA/PA66 composites cannot provide satisfied bioactivity. Therefore, we designed hydrogel with bFGF as a drug carrier to enhance the biological activity of nHA/PA66, not only mainly the ability of BMSCs to bone formation, but also to reduce the degradation of bFGF. The sustained release of bFGF displayed positive effects, solving the problem of the limited duration of the biological activity of bFGF [15].

The results of Alizarin Red S and ALP staining showed the same tendency. These two bFGF groups both showed significant higher osteogenesis abilities than others. At the same time, nHA/PA66/D-RADA16/bFGF exhibited better osteogenesis abilities than nHA/PA66/bFGF. We can conclude that bFGF would be released at the late stage and produce persistent effects. In future studies, it may be possible to adjust the release of bFGF by varying the amount of bFGF contained within the material. Based on current studies, the combination of bFGF may have broad applications in bone regeneration medicine. It has been reported that the combination of bFGF could induce stem cells' osteogenic

differentiation [42,43]. The current application of bFGF can be summarized into three types. Firstly, apply directly to promote endogenous stem cell proliferation and differentiation [43]. Secondly, transfect stem cells with an adenoviral vector carrying bFGF or BMP-2 to the defective area to continuously produce the proteins [44]. Thirdly, implant biological materials that loaded the transfected stem cells with the overexpression of bFGF genes into the bone defect area [45]. However, the above application modalities of bFGF released the two factors at the same time, which is not appropriate for the real state of tissue regeneration. Therefore, smarter ways should be developed to release bFGF not only in the early stage of wound healing to engraft stem cells to the defect site, but also bFGF should be sustained release at the late stage to enhance osteogenic differentiation of stem cells and promote bone regeneration.

In vivo experiments, we observed that nHA/PA66/bFGF material can slightly accelerate the healing of defects compared with negative control. Furthermore, two materials (nHA/PA66 and nHA/PA66/D-RADA16) with bFGF were used to repair bone defects of the femoral condyle to determine whether the sustained release of bFGF improved osteogenesis. Micro-CT analysis of the bone tissue (BV, BV/TV, Tb.Th, Tb.Sp and Tb.N) and H&E staining revealed that the nHA/PA66/D-RADA16/bFGF scaffold material exerted a beneficial effect on the healing of bone defects in SD rats (figure 4). Moreover, the nHA/PA66/D-RADA16/bFGF group also displayed improved bone healing compared with the non-sustained-release bFGF group. It is necessary to combine the D-RADA16 hydrogel with other materials to enhance the peptide stability for drug storage and release. bFGF induces differentiation and proliferation mainly by binding to the corresponding receptors. A previous study demonstrated that bFGF led to a time-dependent increase in cell growth and osteogenesis and growth was inhibited at high concentrations of bFGF and osteogenesis was unaffected, while bFGF increased BMSCs matrix proteoglycan and collagen synthesis [46], which could explain that nHA/PA66/D-RADA16/bFGF has better osteogenic capacity than other groups. Furthermore, the animal experiments indicated that the new materials promoted bone healing significantly ($p < 0.05$), which is consistent with the results of Alizarin Red S staining and ALP staining ($p < 0.05$). In the in vivo experiments, where the role of bFGF is complex and multifaceted, we speculate that one of the key functions of bFGF during the process of bone repair is inducing the generation of a functional microvascular network in the early stages of fracture healing, which provides oxygen and nutrients for fracture repair and bone development. More experiments are therefore needed to clarify the underlying mechanisms. Furthermore, generating more new bone means it can indirectly reflect the increased secretion of osteogenic EMC, such as bone morphogenetic proteins, fibronectin, laminin, vitronectin etc. However, the more direct and deeper relationship between EMC and the bFGF sustained-release system still needs further experimental research.

# 5. Conclusion

In this work, the controlled release of bFGF from self-assembling D-RADA16 peptide hydrogels with porous nHA/PA66 scaffolds was examined alongside the bioactivity of the new controlled release materials, including osteogenesis, proliferation. The obtained results indicate that D-RADA16-coated nHA/PA66 composites permit more stable sustained release of bFGF, which may accelerate bone healing and help maintain a more desirable local concentration of bFGF. We believe that D-RADA16-coated porous materials could play an increasingly prominent role in the preparation of composite scaffolds allowing the slow release of signalling molecules, especially for the regeneration of large-area and severe bone defects. In addition, nHA/PA66/D-RADA16 materials may be excellent sustained-release carriers for the treatment of bone tuberculosis or osteomyelitis.

Ethics. All experimental protocols were approved by the Ethics Committee of Chongqing Medical University (IACUC no. 2016-059).

Data accessibility. Data are available within the Dryad Digital Repository: https://doi.org/10.5061/dryad.td2p55f [47].

Authors' contributions. W.Z., B.Q. and D.J. conceived and designed the study. A.Z., X.C., K.L. and S.C. performed the experiments. Y.L. reviewed and edited the manuscript. All authors read and approved the manuscript.

Competing interests. The authors declare no competing interests.

Acknowledgements. The authors thank Hu Buwei from UCLA for his great help in language revision.

Funding. The synthesis and purchase of materials were supported by the National Natural Science Foundation of China (grant no. 81472057 to B.Q.). The animal experiments were supported by the Natural Science Foundation of Chongqing Province (grant no. KJZH17110 to D.J.). The Applied Basic Research Programs of Science and Technology Department of Sichuan Province (grant no. 2018JY0250).

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
