## [Reviewer comments · Royal Society Open Science]

Review History

Decision letter (RSOS-191317.R0)

28-Sep-2019

Dear Dr zhao:

Manuscript ID RSOS-191317 entitled "Controlled release of basic fibroblast growth factor from a peptide biomaterial for bone regeneration" which you submitted to Royal Society Open Science, has been reviewed. The comments from reviewers are included at the bottom of this letter.

In view of the criticisms of the editors, the manuscript has been rejected in its current form. However, a new manuscript may be submitted which takes into consideration these comments.

Please note that resubmitting your manuscript does not guarantee eventual acceptance, and that your resubmission will be subject to peer review before a decision is made.

Once you have revised your manuscript, go to <https://mc.manuscriptcentral.com/rsos> and login to your Author Center. Click on "Manuscripts with Decisions," and then click on "Create a

Resubmission" located next to the manuscript number. Then, follow the steps for resubmitting your manuscript.

Your resubmitted manuscript should be submitted by 27-Mar-2020. If you are unable to submit by this date please contact the Editorial Office.

Editor comments to the Author:

The authors try an interesting new approach to bone defect healing. However, the study seems to be lacking a negative control, and the write-up has discussion mixed in with the results. I would welcome a reworked version of the manuscript that shows negative controls (that is, compares results of the treatment cases to what happens when the animal is left alone; these data almost certainly exist somewhere for this model, and the authors likely need not perform new experiments). In this reworked manuscript, please be careful to ensure that every statement in the results is an actual result rather than an interpretation, as this journal is very strict about ensuring that every statement in the results section is backed up by experiment. More speculative interpretations could be included in a discussion section. Minor notes: please be careful about the number of "significant digits" in your results, and please check the language in the figure captions carefully.

Author's Response to Decision Letter for (RSOS-191317.R0)

See Appendix A.

RSOS-191830.R0

Review form: Reviewer 1

Is the manuscript scientifically sound in its present form?

Yes

Are the interpretations and conclusions justified by the results?

Yes

Is the language acceptable?

Yes

Do you have any ethical concerns with this paper?

No

Have you any concerns about statistical analyses in this paper?

No

Recommendation?

Accept with minor revision (please list in comments)

Comments to the Author(s)

The work entitled 'Controlled release of basic fibroblast growth factor from a peptide biomaterial for bone regeneration' is well conducted and needs some minor revisions, as commented below:

1. Pls analyze other cell morphological factors (other than cell roundness) that are related with cell properties.
2. Reason how more proliferative cells are better in osteogenic differentiation.
3. It is better to analyze ECMs of in vivo tissue samples to see the effect of bFGF release more clearly.

4. Other approaches of novel scaffolds and stem cell therapies (apart from delivery approach) should also be cited to feedback the recent research trend in bone regeneration (some representative review papers helpful for this are shown below)

- de Grado, Gabriel Fernandez; Keller, Laetitia; Idoux-Gillet, Ysia. Bone substitutes: a review of their characteristics, clinical use, and perspectives for large bone defects management. *J. Tissue Eng.* 2018; 9: 2041731418776819; Winkler et al., A review of biomaterials in bone defect healing, remaining shortcomings and future opportunities for bone tissue engineering, *Bone Joint Res.*, 2018, 7(3): 232–243; Leyendecker Junior, Alessandro et al. The use of human dental pulp stem cells for in vivo bone tissue engineering: A systematic review. *J. Tissue Eng.* 2018; 9: 2041731417752766.

Review form: Reviewer 2

Is the manuscript scientifically sound in its present form?

No

Are the interpretations and conclusions justified by the results?

No

Is the language acceptable?

No

Do you have any ethical concerns with this paper?

No

Have you any concerns about statistical analyses in this paper?

Yes

Recommendation?

Major revision is needed (please make suggestions in comments)

Comments to the Author(s)

The manuscript contains interesting analysis and data about the combination of commonly used materials (NHA/PA66) with self-assembling peptides, in the field of bone defects regeneration. The developed experiments and the obtained results are promising at all levels: material characterization, in vitro cell analysis and animal models. However, the text is not easy to follow and in general it does not explain properly the realized experiments and the obtained results. Moreover, the discussion and conclusions do not properly encompass the results of the experiments.

Specific considerations:

1- Material and methods

- Need to be better described to allow the reader following the experiment or even repeat it. In its current form it is not possible.
- It is stated that during the electron microscopy assay, only the surface of the composites are analysed. It is not what it can be seen in the results.
- I miss the supplier of the reagents
- Control and samples groups need to be clearly defined for each experiment
- Cell roundness analysis needs to be explained. Why is it measured using DAPI staining? It is not cell roundness, it is nuclei roundness,
- I miss the passage of the BMSC used for experimentation and the time from cell seeding until the beginning of the experiment.
- Why is cell proliferation analyzed in well plate? The control group should be the composite without RAD16 and/or bFGF.
- I miss an explanation of statistical analysis for each experiment (N,n and statistical analysis).

2- Results

- Material Characterization  needs to be better described in the results
- Cellular deformation? Do you mean cell phenotype?
- Figure 2: control group needs a better picture. Which is the control group in this case: 2D well plate or NHA/PA66?
- Cell proliferation, alzarin red and ALP activity needs to be normalized and expressed as % compared to control groups
- Consider rewriting figure captions.
- I miss an explanation of bFGF interaction with the composite. Is it embedded?
- In vitro osteogenesis results could be better described.

3- Discussion

- Proliferation is not explained.
- The release of bFGF can be adjusted by varying RAD16 concentration, but never varying the amount of bFGF.
- The stability of RAD16 is improved due to it is protected by the macroporus of the NHA/PA66.
- Consider revising avoiding subsections and describing the work as a whole.
- I miss an explanation of the impact of in vivo results.

4- General considerations

- It is important to maintain the nomenclature during all the manuscript.
- As a recommendation, I would recommend to revise the paragraph describing the self assembling peptide RAD16 in the introduction.
- A brief description of the mechanical properties of the used materials should be added since the final application of the composite is bone regeneration.
- A description of the abbreviations is needed.
- To follow the text easily, I recommend to maintain the titles of the subsections of materials and methods and the ones in the results.
- Mechanical assay results should be included in the discussion and compared with previous bibliography.

Decision letter (RSOS-191830.R0)

18-Dec-2019

Dear Dr zhao,

The Subject Editor assigned to your paper ("Controlled release of basic fibroblast growth factor

from a peptide biomaterial for bone regeneration") has now received comments from reviewers. We would like you to revise your paper in accordance with the referee and Associate Editor suggestions which can be found below (not including confidential reports to the Editor). Please note this decision does not guarantee eventual acceptance.

Please submit a copy of your revised paper before 10-Jan-2020. Please note that the revision deadline will expire at 00.00am on this date. If we do not hear from you within this time then it will be assumed that the paper has been withdrawn. In exceptional circumstances, extensions may be possible if agreed with the Editorial Office in advance. We do not allow multiple rounds of revision so we urge you to make every effort to fully address all of the comments at this stage. If deemed necessary by the Editors, your manuscript will be sent back to one or more of the original reviewers for assessment. If the original reviewers are not available we may invite new reviewers.

When submitting your revised manuscript, you must respond to the comments made by the referees and upload a file "Response to Referees" in "Section 6 - File Upload". Please use this to document how you have responded to each of the comments, and the adjustments you have made. In order to expedite the processing of the revised manuscript, please be as specific as possible in your response.

- Ethics statement

- Data accessibility

If you wish to submit your supporting data or code to Dryad (<http://datadryad.org/>), or modify your current submission to dryad, please use the following link:
<http://datadryad.org/submit?journalID=RSOS&manu=RSOS-191830>

- Competing interests

- Authors' contributions

- Acknowledgements

- Funding statement

on behalf of Professor Guy Genin (Associate Editor) and Malcolm White (Subject Editor)
openscience@royalsociety.org

Associate Editor Comments to Author (Professor Guy Genin):

Associate Editor

Comments to the Author:

The paper is much improved since the original submission. However, several aspects of the manuscript still need to be brought into a more standard form. Both reviewers note that many required aspects of the manuscript are missing (for example, the suppliers for reagents), and both note as well that extra care must be made to remove any conclusions that are not specifically supported by the results. Finally, the manuscript could benefit from a thorough rewriting that streamlines the authors' arguments and places these in the context of the current literature.

In addition to strengthening these aspects of the write-up, which in the associate editor's opinion is a requirement for publication in RSOS, Reviewer #1 has some very nice ideas for further analyzing your data. The associate editor recommends considering these carefully, but notes as well that following these suggestions for additional plots is not a requirement for publication.

Reviewer comments to Author:

Reviewer: 1

Comments to the Author(s)

The work entitled 'Controlled release of basic fibroblast growth factor from a peptide biomaterial for bone regeneration' is well conducted and needs some minor revisions, as commented below:

1. Pls analyze other cell morphological factors (other than cell roundness) that are related with cell properties.
2. Reason how more proliferative cells are better in osteogenic differentiation.
3. It is better to analyze ECMs of in vivo tissue samples to see the effect of bFGF release more clearly.
4. Other approaches of novel scaffolds and stem cell therapies (apart from delivery approach) should also be cited to feedback the recent research trend in bone regeneration (some representative review papers helpful for this are shown below)
 - de Grado, Gabriel Fernandez; Keller, Laetitia; Idoux-Gillet, Ysia. Bone substitutes: a review of their characteristics, clinical use, and perspectives for large bone defects management. *J. Tissue Eng.* 2018; 9: 2041731418776819; Winkler et al., A review of biomaterials in bone defect healing, remaining shortcomings and future opportunities for bone tissue engineering, *Bone Joint Res.*, 2018, 7(3): 232–243; Leyendecker Junior, Alessander et al. The use of human dental pulp stem cells for in vivo bone tissue engineering: A systematic review. *J. Tissue Eng.* 2018; 9: 2041731417752766.

Reviewer: 2

Comments to the Author(s)

The manuscript contains interesting analysis and data about the combination of commonly used materials (NHA/PA66) with self-assembling peptides, in the field of bone defects regeneration. The developed experiments and the obtained results are promising at all levels: material characterization, in vitro cell analysis and animal models. However, the text is not easy to follow and in general it does not explain properly the realized experiments and the obtained results. Moreover, the discussion and conclusions do not properly encompass the results of the experiments.

Specific considerations:

1- Material and methods

- Need to be better described to allow the reader following the experiment or even repeat it. In its current form it is not possible.
- It is stated that during the electron microscopy assay, only the surface of the composites are analysed. It is not what it can be seen in the results.
- I miss the supplier of the reagents
- Control and samples groups need to be clearly defined for each experiment
- Cell roundness analysis needs to be explained. Why is it measured using DAPI staining? It is not cell roundness, it is nuclei roundness,
- I miss the passage of the BMSC used for experimentation and the time from cell seeding until the beginning of the experiment.
- Why is cell proliferation analyzed in well plate? The control group should be the composite without RAD16 and/or bFGF.
- I miss an explanation of statistical analysis for each experiment (N,n and statistical analysis).

2- Results

- Material Characterization  needs to be better described in the results
- Cellular deformation? Do you mean cell phenotype?
- Figure 2: control group needs a better picture. Which is the control group in this case: 2D well plate or NHA/PA66?

- Cell proliferation, alzarin red and ALP activity needs to be normalized and expressed as % compared to control groups
- Consider rewriting figure captions.
- I miss an explanation of bFGF interaction with the composite. Is it embedded?
- In vitro osteogenesis results could be better described.

3- Discussion

- Proliferation is not explained.
- The release of bFGF can be adjusted by varying RAD16 concentration, but never varying the amount of bFGF.
- The stability of RAD16 is improved due to it is protected by the macroporus of the NHA/PA66.
- Consider revising avoiding subsections and describing the work as a whole.
- I miss an explanation of the impact of in vivo results.

4- General considerations

- It is important to maintain the nomenclature during all the manuscript.
- As a recommendation, I would recommend to revise the paragraph describing the self assembling peptide RAD16 in the introduction.
- A brief description of the mechanical properties of the used materials should be added since the final application of the composite is bone regeneration.
- A description of the abbreviations is needed.
- To follow the text easily, I recommend to maintain the titles of the subsections of materials and methods and the ones in the results.
- Mechanical assay results should be included in the discussion and compared with previous bibliography.

Author's Response to Decision Letter for (RSOS-191830.R0)

See Appendix B.

RSOS-191830.R1 (Revision)

Review form: Reviewer 2

Is the manuscript scientifically sound in its present form?

Yes

Are the interpretations and conclusions justified by the results?

Yes

Is the language acceptable?

Yes

Do you have any ethical concerns with this paper?

No

Have you any concerns about statistical analyses in this paper?

No

Recommendation?

Accept as is

Comments to the Author(s)

The authors have answer all the questions presented in previous revision and have include the recommendations in the text. I consider that the the text can be published in the current form.

Decision letter (RSOS-191830.R1)

12-Feb-2020

Dear Dr Zhao,

It is a pleasure to accept your manuscript entitled "Controlled release of basic fibroblast growth factor from a peptide biomaterial for bone regeneration" in its current form for publication in Royal Society Open Science. The comments of the reviewers who reviewed your manuscript are included at the foot of this letter.

Please ensure that you send to the editorial office the following:

- an editable, clean version of your accepted manuscript;
- individual files for each figure and table included in your manuscript. You can send these in a zip folder if more convenient.

Failure to provide these files may delay the processing of your proof. You may disregard this request if you have already provided these files to the editorial office.

Kind regards,
Lianne Parkhouse
Editorial Coordinator
Royal Society Open Science
openscience@royalsociety.org

on behalf of Professor Guy Genin (Associate Editor) and Malcolm White (Subject Editor)
openscience@royalsociety.org

Associate Editor Comments to Author (Professor Guy Genin):

Many thanks to you for submitting this excellent paper. Please do continue sending your best work to this journal!

Reviewer comments to Author:

Reviewer: 2

Comments to the Author(s)

The authors have answer all the questions presented in previous revision and have include the recommendations in the text. I consider that the the text can be published in the current form.

Appendix A

Editor comments to the Author:

The authors try an interesting new approach to bone defect healing. However, the study seems to be lacking a negative control, and the write-up has discussion mixed in with the results. I would welcome a reworked version of the manuscript that shows negative controls (that is, compares results of the treatment cases to what happens when the animal is left alone; these data almost certainly exist somewhere for this model, and the authors likely need not perform new experiments). In this reworked manuscript, please be careful to ensure that every statement in the results is an actual result rather than an interpretation, as this journal is very strict about ensuring that every statement in the results section is backed up by experiment. More speculative interpretations could be included in a discussion section. Minor notes: please be careful about the number of "significant digits" in your results, and please check the language in the figure captions carefully.

Respond:

Dear editor :

Thanks for your comments.

A negative control (black control) was added in the manuscript
speculative interpretations have been removed from "Results"

best,

Weikang

Appendix B

Reviewer: 1

- 1. Pls analyze other cell morphological factors (other than cell roundness) that are related with cell properties.**

Answer: I have added other cell morphological factors in the discussion.

Cell morphology was observed by immunofluorescence staining, we can conclude that bFGF affects early cell adhesion. After being cocultured with different groups for 12 h, the morphology of BMSCs in these groups with bFGF was flat and the area large, indicating that the cell spreading is good. Many pseudopodium were observed in the cells, indicating that the cell extending is good, while the cells in the groups without bFGF were smaller with fewer pseudopods.

- 2. Reason how more proliferative cells are better in osteogenic differentiation.**

Answer: No direct results from this experiment suggest that proliferation can promote BMSCs osteogenic differentiation. bFGF promotes cell adhesion, proliferation, and osteogenic differentiation have been reported. But bFGF's rapid inactivation restricts its bioactivity. The main purpose of this experiment is to verify the slow-release ability of this slow-release material in the future, so that the working time of bFGF will be extended. The long-term retention of bFGF may be the reason that the cells of the nHA / PA66 / D-RADA16 / bFGF group have better proliferation and osteogenesis

- 3. It is better to analyze ECMs of in vivo tissue samples to see the effect of bFGF release more clearly.**

Answer: In this study, the u-CT results showed that the bone density of the bFGF controlled-

release group was significantly increased compared with the other two groups. Since the formation of new bone is closely related to EMC, this result can indirectly reflect the correlation between bFGF and EMC in osteogenesis. ECM is a very broad concept that plays a major role in bone formation. ECM contains bone morphogenetic proteins, fibronectin, laminin, vitronectin, etc. The detection of EMC secretion can indeed make the research deeper explore the relationship between ECM and bFGF, and we will conduct further research in subsequent experiments.

4. Other approaches of novel scaffolds and stem cell therapies (apart from delivery approach) should also be cited to feedback the recent research trend in bone regeneration (some representative review papers helpful for this are shown below)

Answer: Reviewers recommend very helpful articles, and I will add all of them in subsequent revisions.

Reviewer: 2

1- Material and methods

- Need to be better described to allow the reader following the experiment or even repeat it. In its current form it is not possible.

Answer: Materials and methods have been Revised

- It is stated that during the electron microscopy assay, only the surface of the composites are analyzed. It is not what it can be seen in the results.

Image-pro plus was employed to measure the macropore size of nHA/PA66 and the length as well

as width of nanofibers (n=12). The porosity of porous nHA/PA66 was measured as follows:

1. Prepare the same volume of nHA / PA66 porous materials and non-porous solid nHA / PA66 material.

2. Measure the mass of each material on the analytical balance to calculate the density of the porous material and non-porous material for the next step of material pores Calculation of rate.

3. The calculation formula of the material porosity is as follows:

$$\text{Porosity of porous bone filling material} = \left(1 - \frac{\text{Porous bone filling material density}}{\text{Non-porous solid material density}}\right) \times 100\%$$

- I miss the supplier of the reagents

Answer: I will add them in subsequent revisions.

- Cell roundness analysis needs to be explained. Why is it measured using DAPI staining? It is not cell roundness, it is nuclei roundness, Cellular deformation? Do you mean cell phenotype?

Answer: I apologize for the inaccurate description. Since DAPI stains the nucleus, phalloidin stains the cytoplasm. A live cell requires both signals. Phalloidin stains were used for the cell roundness test

Cell roundness is employed to measure the cell shape. There are reports that after the cells adhere, the cells will spread slowly to increase the attachment strength. This experiment measured cell roundness and analyzed cytoskeleton morphology to indirectly reflect cell adhesion in the early stage of co-culture.

- I miss the passage of the BMSC used for experimentation and the time from cell seeding until the beginning of the experiment.

Answer: Thanks for the suggestions, I will modify it in the paper. I have added “All cells used in experiments are within 5 passages” in the section of cell culture. I have added the culture time in each experiment. (such as 7 days for ALP activity test)

- Why is cell proliferation analyzed in well plate? The control group should be the composite without RAD16 and/or bFGF.

Answer: I apologize for the inaccuracy. All experiments were co-cultured with different materials in well plates. In this experiment, CCK8 and Edu were used to evaluate the cell proliferation ability. CCK8 uses a method of measuring absorbance to indirectly detect cell proliferation ability. Edu staining statistics rely on the method of staining proliferating cells to reflect cell proliferation ability. The control group used a glass circle. We have previously verified that there are no significant differences between the results of porous nHA/PA66 and control groups.

- I miss an explanation of statistical analysis for each experiment (N,n and statistical analysis).

Answer: Thanks for your question I have added the N and statistical analysis in the manuscript

2- Results

- Material Characterization  needs to be better described in the results

Revise in Material characterization

Answer: TEM analysis demonstrated that D16 successfully self-assembled to form network structures composed of interwoven nanofibers (Fig. 1A). Results of TEM showed that the D16 peptide nanofibers were 10–36 nm in width and 147–602 nm in length, with a pore size ranging from 144 to 255 μm . Results of SEM showed that the NHA/PA66 macropore size was approximately $635.68 \pm 122.50 \mu\text{m}$ with a porosity of $75.47 \pm 5.31\%$ (Figs. 1B, C). As results of nHA / PA66 / D16 / bFGF SEM exhibited , we found that D16 forms a network structure and is distributed on the material surface, especially it can aggregate in the void structure of nHA/PA66 material. These network structures composed the base of drug release. At the same time, the results of compressive strength showed that none of the physical parameters of nHA/PA66/D-RADA16 were significantly different from those of the nHA/PA66 scaffolds ($P > 0.05$). (supplement table 1)

- Figure 2: control group needs a better picture. Which is the control group in this case: 2D well plate or NHA/PA66?

- Control and samples groups need to be clearly defined for each experiment -

Answer: A better detailed description has been revised in manuscript.

Glass circles are designed for the control group

The main purpose of this experiment is to test the sustained release ability of nHA/PA66/D-RADA16/bFGF materials. nHA/PA66/D-RADA16 was used to compare the effect with or without bFGF and nHA/PA66/bFGF was used to compare the control release effect.

- Cell proliferation, Alizarin red and ALP activity needs to be normalized and expressed as % compared to control groups

Answer: Thanks for your suggestion. These results have been normalized to control groups and put in revised manuscript

- Consider rewriting figure captions.

Answer: It has been revised

- I miss an explanation of bFGF interaction with the composite. Is it embedded?

Answer: A better detailed description has been revised in manuscript.

Lyophilized peptide powder was dissolved in deionized water to obtain a 10 mg/ml (w/v) peptide stock solution at 25 °C. Peptide hydrogels in each group were mixed well at 25 °C. In group one,

combining 10 mg/ml (w/v) d-RADA16 peptide solution with PBS (pH 7.4) by the volume ratio of 1:1, and with the stimulation of ionic solution PBS, 5 mg/ml d-RADA16 peptide solution would undergo self-assembly for 24 h in order to produce d-RADA16 peptide hydrogel. While in the bFGF group, 50 µg/ml (w/v) bFGF solution was prepared by dissolving bFGF powder into PBS (pH 7.4). And then 10 mg/ml (w/v) d-RADA16 peptide solution was combined well with 50 µg/ml (w/v) bFGF by the volume ratio of 1:1. As a result, bFGF was embedded in D-RADA16 nanofibers.

- In vitro osteogenesis results could be better described.

It has been revised in results

Includes description of 3D images and histological pictures, and quantitative analysis of uCT

3- Discussion

- Proliferation is not explained.

Answer: Add more explanation of proliferation

The CCK-8 results indicated that there were no significant differences between each group at 6 h.

As for later time points, bFGF could promote cell proliferation in nHA/PA66/bFGF and nHA/PA66/D-RADA16/bFGF groups (Fig. 2F). Specifically, the cell proliferation in nHA/PA66/bFGF and nHA/PA66/D-RADA16/bFGF groups were significantly higher than groups (control and nHA/PA66/D-RADA16) without bFGF at 12 h and 24 h. This effect was more persistent in the nHA/PA66/D-RADA16/bFGF group than in the nHA/PA66/bFGF group, and cell proliferation remained significantly increased after 72 h in the former case. We can conclude that bFGF-

containing materials can significantly promote cell proliferation, while slow-release bFGF materials can extend the effect of cell proliferation to a minimum of 72h.

- The release of bFGF can be adjusted by varying RAD16 concentration, but never varying the amount of bFGF.

Answer: Thanks for this very good suggestion, I will add this in my discussing.

In this experiment, we mainly studied the adequacy of this slow-release system and have not thoroughly explored the effects of different concentrations of sustained-release bFGF on cell proliferation, adhesion, and differentiation at various time points. The role of bFGF is very important, including the effects on blood vessels, immune system, etc. We will study in the next experiment.

- The stability of RAD16 is improved due to it is protected by the macroporus of the NHA/PA66.

Answer: Yes, compared with exposed peptides, D-RADA16 nanofibers agglomerated in the gap can have a longer degradation time

I will add more about this in discussing.

- Consider revising avoiding subsections and describing the work as a whole.

Answer: Thanks for the suggestions, I have revised it in the paper.

- I miss an explanation of the impact of in vivo results.

Answer: Thanks for the suggestions, I have revised it in the paper.

4- General considerations

- It is important to maintain the nomenclature during all the manuscript.

Answer: Thanks for the suggestions, I will revise it in the paper

- As a recommendation, I would recommend revising the paragraph describing the self-assembling peptide RAD16 in the introduction.

Answer: Thanks for the suggestions, I will revise it in the paper

- A brief description of the mechanical properties of the used materials should be added since the final application of the composite is bone regeneration.

A better detailed description has been revised in manuscript.

In the field of bone tissue replacement and repair, biomaterials composed of bioactive inorganic compounds (such as nHA) and macromolecular compounds (such as PA66) can provide better mechanical properties and ductility. Specifically, nHA is the main mineral component of bones. Therefore, HA is extremely biocompatible and does not promote inflammatory responses. Furthermore, PA66 mimics the collagen component in bones, making the material more ductile.

- A description of the abbreviations is needed.

Answer: I have added the description of the abbreviations in supplement.

- To follow the text easily, I recommend maintaining the titles of the subsections of materials and methods and the ones in the results.

Answer: Thanks for the suggestions, I will modify it in the paper

- Mechanical assay results should be included in the discussion and compared with previous bibliography.

Answer: Thanks for the suggestions, I will modify it in the paper

Sever previous bibliographies have been compared in discussion